# Reviewing the potentials of MMOGs as research environments: A case study from the strategy game Travian

**Siegfried Müller** *, **Raji Ghawi, Jürgen Pfeffer**

School of Social Sciences and Technology, Technical University of Munich, Munich, Germany

* siegfried.mueller@tum.de

**Data Availability Statement:** All relevant data are within the manuscript and its Supporting information files.

**Funding:** The author(s) received no specific funding for this work.

## Abstract

Massively Multiplayer Online Games (MMOGs) provide many opportunities for scientists. Previous research ranges from personality trait prediction to alternative cancer treatments. However, there is an ongoing debate on whether these virtual worlds are able to represent real world scenarios. The mapping of online and offline findings is key to answering this question. Our work contributes to this discussion by providing an overview of the findings from network-based team and leadership research and by matching them with concrete results from our MMOG case study. One major finding is that team size matters. We show that high diversity in the type of teams is a major challenge, especially when combined with the immense amount of data in MMOGs. In our work, we discuss these issues and show that a well-grounded understanding of the data and the game environment makes it possible to overcome these limitations. Besides the team size, the aggregation periods play an important role. Regarding MMOGs as research environments, we show that it is important to pay close attention to the specific game-related contexts, the incentive structures, and the downside risks. Methodologically, we apply support and communication networks to show the influence of certain group-based measures (e.g., density, transitivity) as well as leadership-centered characteristics (e.g., k-core, group centrality, betweenness centralization) on team performance. Apart from our findings on centralization in communication networks, we are able to demonstrate that our results confirm the theoretical predictions which suggest that the behavioral patterns observed in MMOG teams are comparable to those observed in offline work teams.

## Introduction

The potential applications of *Virtual Worlds* as research environments are broad. *Massively Multiplayer Online Games* (MMOGs), especially, are often mentioned in this context [1–6]. This is not by chance. The opportunities they offer are quite tempting. "They free the researcher from the burden of data collection and take advantage of large-scale databases and the computational power of virtual worlds to provide huge datasets that can be generalized to the real world" [7]. MMOGs are highly engaging and psychologically meaningful to players,

**Competing interests:** The authors have declared that no competing interests exist.

and few incentives are needed to motivate them [6]. In this context, Williams claims that the practical advantages of game spaces over real-world laboratories can be immense [8]. Castronova believes that virtual worlds might be the modern equivalent of supercolliders for social scientists, and that they deserve to be the next area to receive a great deal of attention [9]. Ultimately, the crucial question is whether the behavioral patterns of people in virtual worlds also correspond to their behavior in the real world. If not, research in those virtual game worlds is purely an end in itself. Results suggest that in fact, many behaviors are similar in online and offline contexts [7, 10–12]. Researchers have approached this question from different angles. Behavioral scientists found interpersonal interaction patterns in virtual spaces tend to resemble those in real-life settings [13–17]. In series of experiments targeting cognition, communication scholars found that people unconsciously respond socially and naturally to media [18]. The same confusion about reality and play appeared in the famous *Standford Prison Experiment*, in which participants behaved as if their role was real [19]. Researchers performed various empirical tests on whether aggregate economic behavior maps from the real to the virtual world [10, 20, 21]. Their results indicated that virtual economic behavior follows real-world patterns. In the area of trust research, Drescher et al. were able to show that an alternative measurement approach to trusting behavior in virtual settings maps with results from traditional offline survey measurements [22]. In the area of network studies, Szell and Thurner tested a series of social-dynamic hypotheses and were able to verify Granovetter's weak-ties hypotheses as well as the phenomenons of network densification, social balance, and triadic closure [23].

Despite the many points in favor of virtual worlds being able to mirror behaviors in the real world, there are some critical points that should be taken seriously. A major criticism of virtual world research is that virtual worlds are simply not real. Therefore, Ross et al. point out that the players in them interact in an artificial environment, and a virtual world is therefore not suitable for providing generalizable information for the real world [7]. A critical example comes from the study of group formation processes. Johnson et al. conducted an experiment in which they were able to draw parallels between team-building processes in (offline) youth street gangs and (online) guilds in *World of Warcraft* [24]. Building on this work, Ahmad et al. applied an adapted model to replicate the above-mentioned experiment in another virtual world *Everquest 2* [25]. The mapping failed. As an explanation, they point out that different online games have different characteristics and nuances, and thus the social dynamics need not always be the same. Another example comes from epidemiology. In September 2005 the creators of *World of Warcraft* created a spell that acted and unintentionally spread like an infectious disease and infected a large percentage of the players [26]. Epidemiologists saw big potentials for learning more about the spread of diseases in the real world [27]. However, it is questionable whether people in the offline world behave in the same way when their lives are at stake. The online player loses, at worst, their statuses acquired in the online world, but in the real world, a person who loses their life has no hope of ever being resurrected [28]. Therefore, Williams concludes that in certain areas there are significant differences between online and offline worlds. Especially in the previous example: "the risks do not map" [8].

For a more systematic approach, Williams introduced a research framework, he calls the *mapping principle* [8]. This framework can be applied to answer the question to what extend the behaviors in one space are consistent in another space. He states: "if there is enough of a parallel between our online and offline worlds, tests of human behaviors in one might be able to tell us something about human behaviors in the other. This is the essence of 'mapping'" [8].

The jury is still out on the ultimately decisive question of whether the observed behaviors in virtual worlds are real [8]. Ross et al. presume that virtual worlds replicate real-world behavior to a considerable degree [7]. Williams' final conclusion is more skeptical. Due to the fact that virtual worlds are very different in the way the environment is shaped and incentives are set,

he states that "not all virtual situations will map to the real. Indeed, it may be that a minority of virtual contexts will map" [8]. Therefore, it is crucial to take context into consideration when conducting research into MMOGs and other virtual worlds. Only when local contexts are well understood and meaningful data is applied, will researchers be able to assure *validity*, meaning "the extent to which an instrument measures what it is intended to measure" [29]. "In virtual world research, the instrument might be the entire world, or some particular portion of it" [8].

Since it is almost impossible to answer the above-mentioned questions in a generally valid way, our research aims to contribute to this discussion by providing a concrete example based on a dataset from the MMOG *Travian* (see https://www.travian.com/international). Based on this case study, we highlight the potentials and the pitfalls, that researchers face when working with these research environments. In particular, we show how hurdles can be overcome and what must be sacrificed to do so. An essential takeaway from this work is that group size is an important aspect to consider early on. Our data clearly shows that groups of different sizes differ significantly in structures and dynamics. A well-thought-out study design must therefore be able to address this feature.

Regarding the theoretical foundation, we proceed as follows. Following Williams' approach of *predictive validity*, we identify well-established social tendencies (e.g., transitivity, group cohesion, or leader-follower interactions) in groups as well as their expected influence on the group's success. We show how network metrics can be applied in the field of research with MMOGs by giving some examples. The hypotheses we derive from the theoretical foundations enable us to conduct the mapping test mentioned above.

Our extended theoretical section aims to give an overview of the current state of literature in the area of network-based group and leadership research. Guided by this theory-driven approach we apply *Social Network Analysis (SNA)* by calculating a set of *network patterns* to verify if the observed dependencies comply with the theoretical predictions from various offline studies. If these findings from the SNA-based literature mirror the social dynamics and performance outcomes observed in our MMOG environment, this is further evidence that the same fundamental social tendencies are present in both worlds.

In some cases, our findings go beyond existing evidence. We hope that these descriptive findings will encourage future research to continue our work. Our intention is to be as transparent as possible by explaining the formative factors of the game design, incentives, and codes of conduct. Therefore, we present an explicit section describing the environment of the game, the scope and limitations of the dataset, the applied filters, and the underlying assumptions. In the final section, we discuss the results. Finally, we will highlight potentials and pitfalls for future work in this still young stream of research into these unusual but also unique environments.

## Theoretical background: Organizations as networks

Researchers have called for organizations to be viewed as networks [30–32]. Networks represent informal structures and have the ability to reveal *what is really going on*. Often, the insights they provide can show, what formal structures fail to explain. "Basic to this approach is the assumption that organizational actors are embedded within a network of relationships" [33]. Interactions between actors are constantly taking place. It is a dynamic process leading to "emergent structures" [34]. New ties form, are renewed - or not. So networks are constantly changing. On the one hand, networks can be regarded as a snapshot representing what was going on in *past interactions* [35, 36]. On the other hand, they represent the environment, which is forming the *future interactions* between the actors. In this context, Burt distinguishes between *opportunities and constraints* [37]. Opportunities reflect the *social capital* of an actor

or group in the form of the options for action. Constraints reflect the limitations an actor or group is facing [37]. Therefore, Borgatti et al. state: "Perhaps the most fundamental axiom in social network research is that a node's position in a network determines in part the opportunities and constraints that it encounters" [38]. Brass et al. add to this: "Interactions which occur within the constraints of structure can gradually modify that structure. For example, those persons disadvantaged by the current structural constraints may actively seek to change them" [39]. This leads to an ongoing adoption process based on the social relationships that are present [40]. Therefore, networks are both, a reflection of the past and the starting point for future interactions.

## Different types of networks (multi-relational networks)

Interactions among team members can take place in many ways. Studies have therefore looked at a broad range of networks. Based on a sample of 492 journal articles from network-based team research over the past 25 years, Grosser et al. found that the most frequently studied forms of interaction were: *communication* (16.5%), *advice* (15.8%), *workflow* (13.3%), and *friendship* (9.0%) [41]. In general, however, the relations can be any type of links between actors, including formal role relations, affective expressions (friendship, respect), social interactions, work flows, transfers of material resources (money, goods), publication and retrieval of knowledge, flows of non-material resources (information, advice), and business alliances, to select just a few [32]. Some forms of interaction have only been studied occasionally. For example, little attention has been paid to the importance of informal network relationships connecting individual actors in terms of dimensions such as support [42]. Interestingly, this type of interaction opens up new ways to study communication processes. Previous research has found that the vast majority of helping interactions are triggered by a request for help which implies that an interdependence with a prior communication tie might exist [43]. This example shows that integrating multiple networks is a very helpful approach in tackling the complexity of real or virtual-world interactions in teams, particularly, because viewing teams as multi-relational teams is closer to the way teams work in natural organizational environments [41]. In the following, we focus on two of the above-mentioned networks types: *communication* and *support*. The idea behind this choice is to focus on ties between actors that are (1) clearly traceable and (2) related to coordination activities that are seen as central to team leadership. Both the communications and support networks meet these requirements. However, they are very different in nature. Communication networks are very useful for tracking the exchange of information between actors and uncovering how influence is exerted within a team. Further, it shows which subgroups are central within the information flow and therefore hold the function of coordinating the group. In contrast, support networks are much more a result of these coordination activities, reflecting the extent of mutual support and cohesion within a team. In addition, they show the extent to which a team is capable of coordinated action.

## Team patterns in communication networks

"Communication has always been viewed as a key element in any group" [44]. Consistent with this, it has been the most frequently studied tie content in network-based team research [41]. Monge and Contractor define communication networks as "the patterns of contact that are created by the flow of messages among communicators" [34]. This implies personal contact, a flow of information, group-decision making, or even the constitution of leadership relationships [45]. Further, the exchange of communication at a distance (such as e-mail, telephone, or messenger) forms the basis for the existence of *virtual organizations* [46].

While formal networks that represent the *traditional hierarchies* within an organization are presumed to represent the channels of communication through which orders are transmitted downward and information is transmitted upward, these networks mostly fail to represent reality. In contrast, informal networks (as communication networks) reflect team dynamics much better by providing information about important changes within the organization like restructuring processes or the change of power [34]. Nevertheless, these informal networks show some restrictions. Cross and Parker note that often, the central people in a communication network are highly involved people like secretaries and office managers who are rarely involved in strategic decisions or leadership functions [47]. Therefore, they highlight that the content of the communication exchange plays an important role. Despite their limitations, communication networks are often applied and can provide valuable insights, especially when combined with other networks.

## Density in communication networks (team cohesion)

The most frequently used measure to map team cohesion is density. "Density is simply the number of ties in the network, expressed as a proportion of the number [of] possible [ties]" [48]. It can also be interpreted as the probability that a tie exists between any pair of randomly chosen nodes. In a directed network, the number of possible edges is: $n(n - 1)$, and we further denote $n$ and $m$ number of nodes and number of edges, respectively. Thus, the density is given by:

$$\text{Density} = \frac{m}{n(n - 1)} \tag{1}$$

One critical aspect to consider is "the fact that the density depends on the size of a graph" [49]. Therefore, density measures should not be used to compare networks of different sizes [50]. Density is very frequently studied in network-based team research. It has been applied as a measurement in 17.9% of the studies [41]. Multiple meta-analyses [36, 51–53] confirmed that teams with densely configured (positive) ties tend to perform better. The literature offers a couple of possible explanations for this finding. Teams with a high density in internal communication show higher levels of information sharing necessary for successful task completion [36]. This includes tacit knowledge or vital job-related ideas low-density teams might be unwilling or unable to exchange [54, 55]. Dense networks further reduce the obstacles to initiate coordinated action [56]. Another aspect is that teams with a low density might have to rely on a few individuals who act as *brokers* between disconnected parts of the team. This restriction in *structural autonomy* may lead to calculated or involuntary bottlenecks in information flow or coordination which might harm task completion and therefore the overall effectiveness of the team [37]. Further, Coleman argues that *closure* (dense connections), which enables effective sanctioning, is a basic requirement for the existence of effective norms [57]. In addition, this *social capital* fosters "the trustworthyness of social structures that allows the proliferation of obligations and expectations" [57]. "The density of relationships also may enhance social consensus on issues, thereby increasing moral intensity, perception, judgment, intent, and action" [33]. With respect to the potential of this social capital, Balkundi and Kilduff argue that, assuming a positive attitude toward the group leader, "a dense network [tends] to enhance [. . .] the leader's effectiveness" in serving the collective good [58]. However, a high internal density makes it less likely that members will seek knowledge from external sources [59]. Combined with the tendency of highly connected networks to share redundant information [37, 60], this can hinder value creation, innovation, and thus future team performance. Nonetheless, there is a broad consensus that high density has a positive impact on team

functioning. Given these theoretical explanations and findings, our hypothesis regarding density in communication networks is:

*H1: Analogous to findings in offline studies, observations from our MMOG setting will show that a higher density in a team's communication networks positively correlates with higher team performance.*

## Transitivity in communication networks (structural holes)

Transitivity has a long history in social network analysis. In the field of communication networks, it often appears in the context of *structural holes theory*. Burt defines a structural hole as the absence of a link (direct or indirect) between two actors [37]. "Sociological theory offers a role describing people who derive control benefits from structural holes. It is the *tertius gaudens*, the third who benefits: a person who derives benefit from brokering relationships between other players" [37]. This actor is called the *broker*. Two possible tertius gaudens strategies exist: First, the broker can play two people against each other (e.g., by withholding important information). "Second, people can be played against one another when they make conflicting demands on the same individual in separate relationships" [37]. These invisible relations cannot be measured directly. They are "visible only by their absence" [37]. One possible tool that can be used to measure the extent this phenomenon is present within a communication network is the amount of *transitivity* within a team. This is simply defined by "the number of transitive triads divided by the number of transitive plus intransitive triads" [48].

$$\text{Transitivity} = \frac{\text{number of closed triads}}{\text{number of all triads (open and closed)}} \tag{2}$$

In calculating transitivity for directed networks, it is typical to just ignore their directed nature, and apply Eq (2) as if the edges were undirected [61].

With the application of this measurement, it is possible to get an overview of the proportion of triads, where the broker has the *opportunity* to exploit their position. Borgatti et al. state: "as always, the numbers are not terrible informative in themselves: they are best used comparatively" [48]. In this context, it should be mentioned that the existence of an opportunity does not necessarily lead to its exploitation for one's own benefit. Therefore, Brass et al. introduce the term of potentially *unethical behavior* [33]. They claim that "opportunities presented by structural holes do not necessarily result in unethical behavior. [. . .] For a person with moderate moral character, the opportunity provided by a structural hole may be the difference between acting ethically or unethically". However, we assume that the more often a structural opportunity exists, the more often the individual brokers will finally misuse the situation to maximize their own benefit instead of the benefit of the group. We, therefore, assume that a high amount of transitivity positively influences team collaboration and performance. Given these theoretical foundations our related hypothesis is:

*H2: Analogous to findings in offline studies, observations from our MMOG setting will show that, higher transitivity in a team's communication networks positively correlates with higher team performance.*

## Leadership patterns in communication networks

Leading a team implies much more than holding a formal position. A widely used definition is that leadership is "a process of social influence through which an individual enlists and mobilizes the aid of others in the attainment of a collective goal" [62]. While researchers largely

agree on the importance of a *collective goal*, the question of whether leadership must be exercised by a single individual is controversial. In this context, Stogdill and Shartle point out that it is critical "to discover what leaders do" rather than to focus on "who they are" [63]. This "signaled a shift in focus from the individual leader to the behavior of individuals in leadership roles" [64]. A central implication from this shift is that leadership roles can be performed by more than one single person [65–67]. This means that not only formal leaders but "all group members could fulfill necessary leadership functions" [64]. When these behaviors are performed by group members, the related structural patterns (e.g., central position) can be found in the intra-group's communication network. Therefore, these informal networks provide rich opportunities to unveil the enactment of leadership in work teams. This is due to the fact that the occurrence of leadership influence and informal ties are unlikely to occur independently of each other [34]. Friedrich et al. state: "Communication is essential to collective leadership." It is "a 'prerequisite' for understanding the problem that the team is facing, defining shared goals, understanding where the relevant expertise lies in the network, and sharing the leadership role" [68]. Other relational leadership theories [69, 70] go so far as to regard leadership "as socially constructed through communication exchanges" [45]. Further, several scholars have shown that certain network patterns (e.g., central positions) are associated with holding a leadership position [71–73]. The basic idea behind this relationship is that "leadership can be understood as social capital that collects around certain individuals—whether formally designated as leaders or not—based on the acuity of their social perceptions and the structure of their social ties" [58]. When a group has more than one central actor (leader), individual-level measures fall short of describing the influence of the (informal) leadership subgroup. The concept of *group centrality* [74] can solve this issue. Unfortunately, there is no broad consensus in the literature on how to delineate such an informal group of leaders. Therefore, we propose to apply the k-core approach [75], which offers a convenient way to identify leadership structures in a group's networks [66, 76]. The affiliation to a "maximally cohesive nucleus" [75] can hereby serve as the attribute needed to define the relevant subgroup. The combination of the concepts of *group centrality* and *k-core* enables us to capture the influence of a leader's centrality on specific outcomes (e.g., team performance) not only at an individual level but also at a group level. In the following section, we will present background information about the mentioned concepts and show how they can be complemented by established network metrics.

### Individual level centrality in communication networks (centralization of influence)

One way to measure leadership influence is *centrality*. "[It] is a proxy for an individual's influence in the social system" [77]. Research has shown that individuals holding a central position are more likely to be seen as (informal) leaders by other team members. In this context, multiple studies have reported a positive relationship between a leader's network centrality with social power and the activation of leadership perception [58, 78, 79]. Individual actor's attributes cannot only be interpreted directly, but their distribution allows additional insights, about how unequal (leadership) influence is exerted within the team. An important measurement, which is often applied in the context of *shared leadership* is *centralization*. This approach goes back to Pearce and Conger's definition that "leadership is broadly distributed among a set of individuals instead of centralized in hands of a single individual who acts in the role of superior" [65]. This view implies that the amount of shared leadership should be seen as a continuum ranging from strictly hierarchical to equally distributed among all team members [69]. To measure this effect, Majo and Pastor propose that "centralization refers to the degree to which all members of the network are unequally central in the [leadership] network" [77]. "To

calculate centralization [see [80]], we sum the difference between each node's centrality and the centrality of the most central node.[...] We then divide this by the maximum possible" [48].

Thus, let $c_1, \cdots, c_n$ be node-level centrality measures, where $c_i$ is the centrality of node $i$ by some metric (e.g., betweenness). Let $c_*$ denote the maximum of those centrality measures: $c_* = \max\{c_1, \cdots, c_n\}$, and let $S$ denote the sum of the difference between $c_*$ and $c_i$ for all nodes: $S = \sum_i [c_* - c_i]$; then $S = 0$ if all nodes are equally central; while $S$ is larger if there is at least one node that is more prominent than the other nodes.

Network centralization is given by:

$$C = \frac{\sum_i [c_* - c_i]}{\max \sum_i [c_* - c_i]} \tag{3}$$

For degree, closeness, and betweenness centrality measures, the most centralized structure is the star graph. A star formation represents hierarchical leadership with a central leader in the middle. The other extreme case is a fully decentralized network in which all actors show exactly the same centrality score. Although this approach is widely accepted in the literature, it has an important limitation: it is designed for the study of explicit *leadership networks*. It is therefore questionable if it can also be applied to other types of networks. Communication, for example, plays an important role in exercising leadership influence, but not all interaction involves leadership. In the case of *communication networks*, a possible solution is the application of the concept of *betweenness centrality* [23, 80–84]. Mullen et al. show in their meta-study that especially betweenness centrality is suitable for predicting leadership [72]. This measure reflects how important (powerful) an actor is as a *broker* and, conversely, how dependent the team is on that person. This is a double-edged sword. On the one hand, the team can benefit from good coordination. On the other hand, this position can also represent a risk. As Balkundi and Harrison mention: "These brokers may engage in calculated or involuntary filtering, distortion, and hoarding of information, hampering the team's eventual task completion" [36]. When it comes to the influence of centralization on performance, "there has been little consistency in the findings pertaining to [general] network centralization and team performance" [41]. Separately, several meta-studies have shown a positive impact of distributed leadership on team performance [85–87]. In our more specialized case of betweenness centrality in communication networks, we assume that a low centralization (high de-centralization) will represent a high level of shared leadership [77] and thus lower dependency on a single central actor who acts as a broker, which results in higher team performance. We, therefore, hypothesize that:

*H3: Lower betweenness-centralization in a team's communication networks indicates higher team performance.*

## K-core in communication networks (subgroup of most prominent actors)

In the last section, we showed that an actor's centrality is highly related to their social power and whether a person is perceived as a leader by other group members. Therefore, we are interested in learning more about the influence the subgroup of these most prominent actors has on team dynamics. When it comes to studying subsets, the most important question is that of delineation. Based on individual centrality scores (degree), the concept of k-core is a very helpful approach for targeting these most prominent actors. This approach goes back to the basic assumption that "all cohesive subsets are contained in k-cores" [75]. This includes the most prominent group of highly connected actors - in most cases, the members are the formal

and informal leaders of the group. Formally, a k-core of a network G is a maximally connected subgraph of G "in which each node is adjacent to at least a minimum number, k, of the other nodes of the subgraph" [88].

The *coreness* (aka. core number) of a node *u*, denoted *coreness(u)*, is the largest value *k* of a *k*-core containing that node. Thus, a node *u* has a *coreness c* if it belongs to a *c*-core but not to any (*c* + 1)-core.

The largest *k* for which *G* has a *k*-core is called the degeneracy of *G*, and denoted $k_{max}$.

$$k_{max} = \max_{u \in G} \{coreness(u)\} \tag{4}$$

The main core of a network is the core with the largest degree,

$$core(G) = \{u \in G: \ coreness(u) = k_{max}\} \tag{5}$$

from which we are able to compute the size of the main core,

$$kcore\ size = |\{u \in G \mid coreness(u) = k_{max}\}| \tag{6}$$

as well as the k-core relative size, which is the fraction of nodes in the k-core in relation to all nodes in the network.

$$kcore\ relative\ size = \frac{1}{N}|\{u \in G \mid coreness(u) = k_{max}\}| \tag{7}$$

The members of such a main core are not only connected with each other, but even more importantly they are also *possibly* connecting, and therefore integrating others outside their (elite) subgroup. In addition to this, Seidman has shown, that communication outside the frontier of a 3-core breaks up into components which are trees. This "tree structure of the components of the frontier can be used to construct a hierarchy" [75], which forms the foundation of a hierarchical leadership structure. Therefore, examining the *k-cores* can help us gain a deeper understanding of the hierarchical composition of a team [76]. Besides the ability to identify the individual members of the most cohesive subgroup, the k-core approach also provides information about the size of this important subgroup (k-core size). This number can now be put in relation to the group size of the entire team. The resulting ratio represents the percentage of team members we have defined as the (informal) group leaders. The higher this ratio, the higher we assume that the level of shared leadership is within the team. This allows us to take an alternative approach to determine the extent of shared leadership. In connection with performance, multiple meta-studies have confirmed that shared leadership has a positive influence on team performance [85–87]. Therefore, our related hypothesis is:

H4: *A higher relative group size of the maximal cohesive subgroup (k-core) in a team's communication networks indicates higher performance.*

## Group level centrality in communication networks (connectedness of the core)

Using a team's leader network *centrality* as a predictor for team success has a clear limitation. It only applies to individual actors. When a group is led by more than one person, this measurement no longer fits. To overcome this limitation, Everett and Borgatti introduced another set of network indices: *group centrality* [74]. When calculating group centrality, the first step is to define the subgroup to target. The criteria that are used to define it can be derived from an *individual attribute* (e.g., degree), as well as from *cohesive subgroup techniques* (e.g., cliques),

or *positional analysis techniques* (e.g., structural equivalence). As introduced above, *k-core* is one such technique for defining such a set of actors, which we regard as the *core* or *leadership subgroup*. Second, the centrality measures need to be adapted to the new context. For the most common centrality measures, Everett and Borgatti have provided proper generalizations that can be applied in our context. An important criterion is, that such a measure needs to yield the same results as the traditional measurement when applied to a group consisting of a single individual. Therefore, the "centrality of a group is computed directly from the network of relationships among individuals", instead of computing centrality based on a network of relationships among groups [74].

In our study, we consider the group of nodes in the main *k*-core, i.e., $C = core(G)$, and we construct the reduced group network. Then, as a feature, we use the degree centrality of the single entity that represents the reduced group.

Thus, given a network *G*, and a group of nodes $C \subseteq V(G)$, the *group reduced network* is a network *H* in which group *C* is reduced to a single entity $u_C$, $V(H) = (V(G) \backslash C) \cup \{u_C\}$ and each other node (in $V(G) \backslash C$) that was connected to any node in group *C*, is connected to $u_C$.

When it comes to interpreting these measures, it can be said that *group centrality* refers to the connection the most prominent subgroup holds with the remaining group. This is a piece of important information, which tells us if this group is able to reach the whole *periphery*. A second measure that is very useful to complement this view is the application of the *maximal k-core*. This value represents the internal connectedness of the *core*. Combining these two measures can therefore be a helpful extension to predict team dynamics and performance. In their meta-analysyis Balkundi and Harrison showed that leaders with a high centrality within a team's intra-group networks tend to perform better [36]. This relationship is based on focusing on the individual level. In our study, we extend this view to a subgroup of (informal) group leaders. Moreover, we assume that this relationship remains unchanged. This means, the more the *subset of core actors* is connected to the rest of the group (*periphery*), the better it can accomplish its coordination task and the more successful a team will be. Furthermore, we assume that the higher the maximum observed k-core value (which necessarily correlates with the size of this subgroup), the better the group's ability to steer the team. Our final hypotheses are:

*H5: A higher degree of group-level centrality in a team's communication networks indicates higher performance.*

*H6: A higher maximal k-core value in a team's communication networks indicates higher performance.*

## Team patterns in support networks

Support networks play an important role in a group's interaction. They unveil both, the outcomes from coordination processes and they disclose the factual flow of resources between team members. This is important in that it represents how limited resources are allocated. The allocation of resources is widely regarded as a central leadership function [89]. Therefore, we expect that studying the factual flow of resources in combination with communication will yield important insights into the leadership structure of a given team and the context in which it is exercised. Despite their importance, support networks have only occasionally been applied in the area of network-based team and leadership research. This might be due to limitations in data collection and/or availability. Another alternative explanation might be that the literature rarely provides any theoretical foundation. So far, no consensus or all-encompassing definition

of what constitutes a support network exists. In the following, we will refer to the term *support* as:

*'Supporting another team member by providing scarce resources, which are needed to accomplish the goals of the recipient or the team as a whole'.*

In the next section, we present three different network metrics based on support networks that play an important role in team functioning.

## Active participation in support networks (isolates)

In a social network, the participation of actors is a simple measure that refers to the fraction of nodes that are connected to other nodes (in contrast to isolated nodes). Let $n$ be the number of all nodes, and $s$ be the number of isolate nodes, then the participation is given by:

$$\text{Participation} = \frac{n - s}{n} \qquad (8)$$

Notice that $n - s$ is the number of active (participating) nodes.

Theories on group members exchanging resources have a long history in research (e.g., [90–92]). One aspect is *prosocial behavior* performed by organizational members with the goal or expectation that it will be for the benefit of the individual or group at which it is directed. Besides some limitations, the general finding is that "prosocial behaviors are very important for organizational effectiveness" [93]. Hence, "research has revealed a number of positive benefits of helping for group and organizational effectiveness" [43]. Therefore, Ng and Van Dyne regard "helping behavior in work groups [. . .] is a critical phenomenon in organizations" [94]. Besides the general importance of the presence of mutual support for the team's performance, another critical aspect is that a high participation rate of team members in supporting each other implies a low amount of isolates within a network. Apart from the negative psychological effects of isolation, being isolated is accompanied by many negative side effects for the group and its functioning [95]. Integrating team members is broadly seen as an important leadership task that is critical for a team's success [46]. Therefore, Balkundi and Kildhuff state: "the extent to which such isolates are part of work groups may predict the extent of leader effectiveness in such groups" and therefore reflecting the effectiveness of the whole team [58]. These two factors: the generally positive effects of support (helping) and the reduced negative effects of the presence of isolated parts, lead us to our next hypothesis:

H7: *Analogous to findings in offline studies, observations from our MMOG setting will show that a higher rate of team member participation in mutual support (fewer isolated individuals) positively correlates with higher team performance.*

## Transitivity in support networks (structural balance)

One traditional approach to thinking about *transitivity* is that "the friends of your friends are your friends" [96]. In regard to helping, this implies that when you provided help to a friend, he or she will most probably also provide help to another friend of yours. Technically, transitivity can be easily measured by dividing the number of transitive (closed) triads by the number of all triads [48]. Eq (2) is used here analogously. We can assume that in a functioning social system, *structural balance* is a "natural tendency" in networks [97]. Therefore, White et al. state that "triadic effects reflect the human propensity to operate in a group structure" [42]. This implies that in a functioning group, we will observe a high amount of closed (transitive) triads compared with the number of intransitive triads [98]. If our observations show low

transitivity, two possible explanations may come into play: First, the team is still in an early phase of self-organization (forming stage) and has not yet reached the later performing phase [99]. Second, a negative influence (e.g. leadership, structural holes) is hindering this "natural tendency" from occuring. In both cases, we assume that the presence of a low amount of *structural balance* will reflect dysfunctional tendencies and therefore is related to lower team performance. This leads us to the following hypothesis:

*H8: Analogous to findings in offline studies, observations from our MMOG setting will show that higher transitivity in a team's support networks positively correlates with higher team performance.*

### Coordinated actions in support networks (in-degree centralization)

Giving direction and coordinating the team is widely seen as a central aspect of leadership [100]. This includes the allocation of (material) resources [89]. We propose *coordinated action* as a way to measure the outcome of executing this function. The group's ability to support a single team mate who is in need of help, especially, can be regarded as an indicator of a high degree of coordination and hence, effective leadership. Therefore, we assume that incoming support concentrated on a central actor might be an indication of a highly effective team. One network measure that can capture this pattern is *in-degree centralization*. Here, this measure is calculated analogously to Eq (3) on the basis of the in-degree. Our assumption is that observing high in-degree centralization indicates a high ability of the team to join forces and to concentrate on the accomplishment of critical actions. This leads us to our third hypothesis regarding support networks:

*H9: Analogous to findings in offline studies, observations from our MMOG setting will show that higher in-degree centralization in a team's support networks positively correlates with higher team performance.*

## Materials and methods

### The virtual world of Travian

Initially published in 2004, *Travian* is a commercial browser-based MMOG. In 2009/10 - the period of data collection - it was operated in 53 countries. To date, more than 150 million players have signed up and participated in one or more rounds. The game is adapted to local markets and is organized in game worlds (servers) where up to 20,000 players compete with each other in real time. Each round lasts approximately one year. The game is highly competitive and can only be won in cooperation. Effective team coordination, in particular, is critical to success. Since only well-managed teams have a chance to be successful, the game Travian is very well suited for a closer examination of the interaction patterns involved. After the start of a game world, players team up with others to form alliances. The size of these groups ranges from 2 to 60 members, a limitation imposed by the game design. Starting with one village, the task of the players is to grow resources, level up their infrastructure, and build armies to protect their kingdom. Armies are not only built for your own protection, they can also be used to fight wars and conquer new territories. The player also has the choice to produce resources on their own or to raid resources from other players. For this purpose, the player has various military units at their disposal, each of which has a stronger offensive or defensive orientation. Besides the development and expansion of one's own villages (territories), one of the most

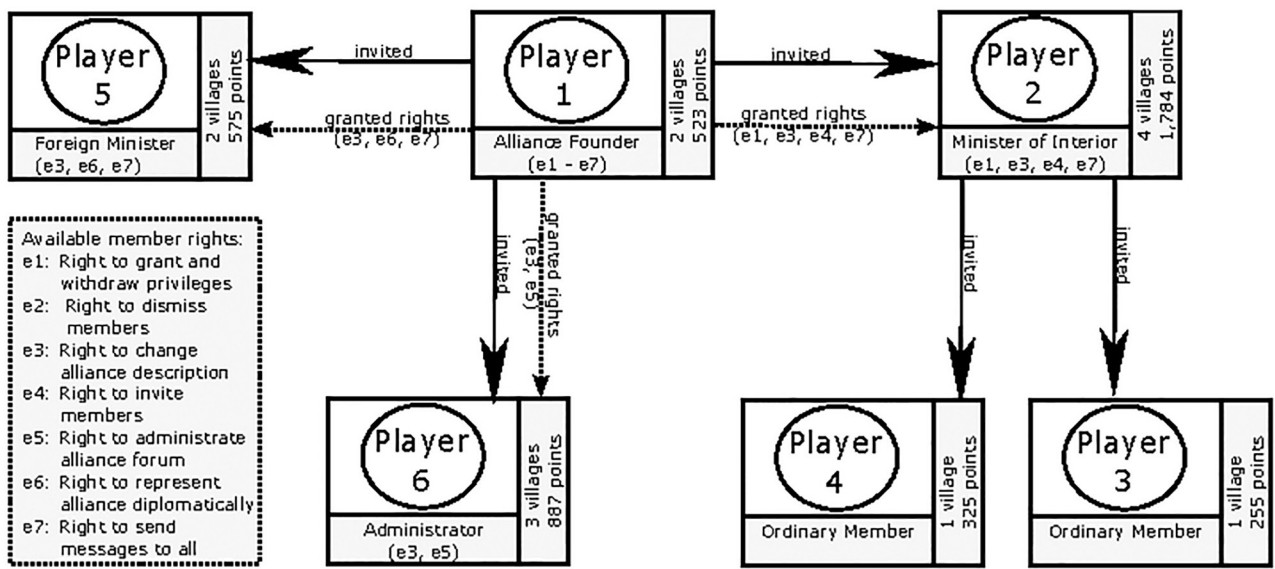

**Fig 1. Teams in Travian are formed by the founder of thealliance, who initially holds all the power.** To grow their alliance, the founder invites players to join the team. Here the founder can grant special rights to the new alliance members. These rights give these players the de facto power to fill different roles within the alliance. As players expand their villages, they earn additional points in the game ranking. The ranking of the alliances derives, in turn from the sum of the points of their individual members.

important aspects of the game is membership of an alliance. These alliances form a certain community of fate in which the members are strongly dependent on each other. Membership in alliances is by invitation only. Therefore, one of the most critical tasks of an alliance founder who also serves as a formal leader is staffing the team (see Fig 1). These alliance leaders are highly dependent on the contribution and commitment of each member. Alliances (especially high-ranking ones) have very strict standards in terms of time investment, individual performance, and contribution to common goals. Individuals who do not meet these requirements are often dismissed or not invited in the first place. Most teams are not led by just one leader. In most cases, management functions are distributed among a group of players acting as an (informal) management group. These functions range from formal authority to admit or dismiss members, to coordinating military operations or mutual support activities, to representing the alliance as ambassadors.

Communication exchange between players takes place via an in-game messaging system (IGM) and, for broader internal information exchange, a forum that can be accessed by alliance members only. Each interaction between players is recorded in the game database in combination with a unique timestamp. In addition, the game records every resource transfer and grant of support (sending reinforcement troops).

The game clearly defines the actual performance of an alliance through a public real-time ranking. This ranking is based on the number of points the alliance members have been able to accumulate since the start of the game round (server). A player gains additional points when successfully upgrading his or her infrastructure. In the event of a lost conflict (battle), the player's and their alliance's points are reduced accordingly. The alliance ranking is based on the sum of the points of all alliance members. Therefore, alliance leaders face a trade-off when it comes to optimizing the ranking position. The easiest way to to so is to add as many members as possible. However, this comes at the cost of losing the efficiency and agility of coordinating a smaller unit. It can therefore be observed that many high-performance teams

voluntarily refrain from exploiting the maximum size of 60 members. However, it must be said that it is crucial to keep group size in mind when evaluating and comparing teams.

Another important environmental factor is time. The ranking position of the player and thus the alliance depends not only on actual growth rates but also on past growth rates. Therefore, when comparing the success of alliances or players, this can be done based on the same day (since the start of the server) in combination with the relative ranking position (e.g., quantiles) of the teams.

## The dataset

The data collection for this research project took place in 2009/10. For a period of approximately one year, the operator of the game, Travian Games GmbH, granted access to their game databases. In total, the company provided 3,294 daily downloads of the game databases from 22 national servers. For privacy protection reasons, these SQL files were *cleaned* beforehand, meaning that all personal data and communication content had been removed before sharing the data. All players were informed by the operator that they were (anonymously) participating in a scientific research project. The participants agreed by accepting the general terms and conditions. Participating in additional surveys was voluntary and incentivized by a small monetary reward. These surveys are not part of this study. The majority of the players originated from Germany, Austria, and Switzerland. Some limited demographics are known from a general survey: 77% of the participants were male, and on average 30.3 years old. They had completed 4.0 years of higher education (e.g., university) on average and 62% were employed. Further information was not available for this study. In total, the data collection period took 356 days. It started with the first day of the server on August 14th, 2009 and ended on the final server day on August 5th, 2010. Eleven out of the 356 data points (SQL files) were missing. In the case of aggregating performance data and team member composition, we interpolated this missing information by using the last known data. In the case of communication and support interactions used to compile the associated networks, we opted to omit this data. In total 82,564 players signed up during the one-year period. Over this period, an average of 8,385 players were active on a daily basis, ranging from 13,318 within the first 100 days of the game round to only 3,078 daily active players in the last 100 days of the server. 74.02% of the players opted to join an alliance at least once. During the above-mentioned period, 4,758 alliances were formed, while 415 alliances still existed at the end of the game round. A total of 3,075,088 messages were sent between players, of which 1,893,756 messages were for communication within an alliance and 1,181,331 were for exchanges between alliances. Furthermore, 2,839,738 support interactions took place between players, i.e., the provision of resources or support troops. 2,088,174 of these interactions occurred outside of their own alliance, while 751,564 occurred between team members.

## Study design

With our exemplary study, we investigated whether certain network-based interaction patterns and their effects on team performance identified in offline-studies can also be found in online environments. To do this, we extracted *intra-team networks* based on communication and support interactions in combination with performance data from a historical dataset of the MMOG Travian as described above. We aim to contribute to the discussion of whether, "virtual worlds are [or can be] suitable environments for drawing conclusions about individual and group behavior in the real world" [7]. In his *mapping research framework*, Williams proposed an approach to how the research community should systematically explore the factors that distinguish MMOG research environments from each other [8]. The process he describes

is intended to enable the research community to identify "what situations, contexts, levels of analysis, and types of human interactions can successfully be tested within virtual spaces, and which cannot" [8]. Our present paper will not be able to answer or discuss this topic in an all-encompassing way. Due to the extensive literature, an assessment of all research environments of the studies (offline and online) mentioned in this paper would be the subject for a separate meta-study and therefore would go beyond the scope of this paper. Our intention was rather to contribute to the discussion by offering detailed insights based on real data from a concrete MMOG-environment. Methodically, we followed the classification from William's comparative framework, who identified four major factors: (1) *directionality*, (2) *group size*, (3) *traditional controls and independent variables*, and (4) *contextual and social architectural factors*. Further, we extended these items by: (5) *type of networks*, (6) *dependent variable*, and (7) *observation period*.

## Directionality

The first question we address is *directionality*. For our *mapping test*, we opted for *'offline to online'*, which means that we identified established findings from the literature that came from studies conducted in traditional offline settings. In the following, we checked whether the relationships identified from the literature were consistent with those we had observed.

## Group size

Overcoming the challenge of working with a *high variety of team sizes* turned out to be a key challenge in our current MMOG research project. This challenge arose for two reasons: the applied aggregation of data points into periods of 60 days each and the specific design of the game. The fact that members can join or leave the alliance at any point in time allows group sizes in our dataset to exceed the upper limit defined by the game. Travian's game design allows team sizes from 2 to 60 members. This means that teams differ in the number of members. Furthermore, the team sizes are not equally distributed; i.e., we have many small teams and fewer large teams in our sample. Generally, it is difficult to compare teams of different sizes for different reasons. Slater found that teams of different sizes show differences in leadership structure, problem-solving, and participation rates [101]. Further, he states that larger groups tend to be less stable, have more difficulties in communication and inhibit member participation. On the other hand, they can be more effective as they have more skills and resources at their disposal. Still, others have shown that large organizations tend to survive longer [102], while smaller groups are more effective at coordination [103].

In addition to these theoretical justifications for treating groups of different sizes differently, the game itself provides an additional argument for doing so. The performance ranking implemented in the game simply adds up the scores of all alliance members and compares them to the scores achieved by the other teams. Therefore, alliances with more members tend to rank higher than smaller alliances. This leads to a very high correlation of 0.757 (Spearman) between team size and ranking position (performance), which can be seen as a very strong bias if this aspect is not treated accordingly. Fig 2 highlights this relationship between group size and team performance. Finding a solution for this was tricky. Such an approach had to allow us to compare teams of similar but different sizes without compromising the analysis due to a very small sample size and the bias mentioned above. Given these limitations, we decided to divide the groups into four subsamples: small groups (15–25 members), medium groups (35–45 members), large groups (55–65 members), and a control sample that includes all team sizes (10–143 members).

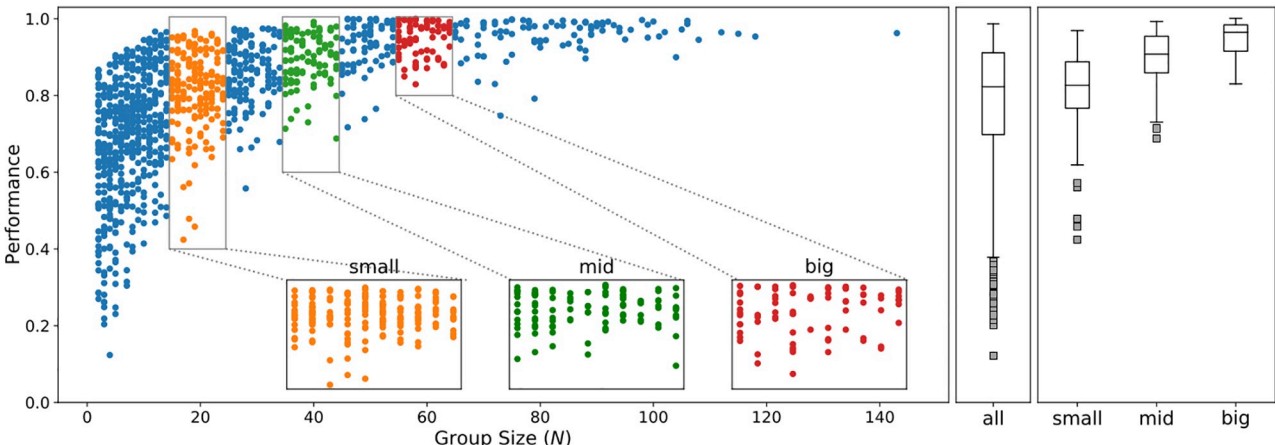

**Fig 2. Splitting the sample of all teams (with 2 to 60 members) into four specific subsamples was a prerequisite for comparing the network patterns of teams of different sizes.**

### Traditional controls and independent variables

In his "research agenda for virtual mapping and other tests," Williams suggests a number of factors that may be relevant to study design as control variables when working with MMOGs [8]. Table 1 addresses these items.

One problem we struggled with was the availability of individual- and team-level data from the game that would be suitable as control variables. As Table 1 shows, apart from network data based on interactions between players, no additional information was available on motivation, psychological profiles, demographics, or communication medium.

### Contextual and social architectural factors

Williams argues that there are a variety of contextual variables that distinguish one virtual world from another [8]. When it comes to drawing conclusions from studies conducted on different (online) environments (e.g., equivalent studies conducted in different MMOGs), it is essential that these contextual factors are known and can thus be taken into account. Following Williams' classification, Table 2 provides an overview of those characteristics that make up the virtual world of Travian and the interactions that take place within it.

### Type of networks

In order to cover a wider range of aspects, we decided to integrate two types of networks, which are very different in nature. *Communication networks* represent the flow of information and reflect coordination activities. *Support networks*, on the other hand, represent the outcome of these activities: the factual flow of resources within the team. The data we needed to construct these networks could be extracted from the game database, where all interactions were stored in combination with a unique timestamp. Thus, data collection can be classified as *unobtrusive*. For communication, we integrated all messages based on personal communication (1:1), but excluded all mass communication (1:n). For support, we grouped two kinds of transfers together: sending raw materials and sending support troops. The game database specifies both the sender and the receiver of a transmission/communication. Therefore, all applied networks were *directed networks*. Finally, it should be mentioned that we chose to construct *unweighted networks* in both cases. The major argument for the decision to use unweighted

**Table 1. The world of Travian—Traditional controls and independent variables based on Williams' research agenda for virtual mapping and other tests.**

| | |
|---|---|
| Motivations | Competition with others is one of the main motivations for players to participate in an online strategy game like Travian. The game participants' primary goal is to achieve a high rank for themselves and the alliance they belong to. In order to pursue this goal, the player must continuously develop their infrastructure and expand their territories. In addition, the design of the game requires a high level of interaction with other players and a significant investment of time by the player. Players in Travian must be willing to keep up with this high level of competition since, otherwise, they will be kicked out of the game by the other players. |
| Psychological profile | The existing dataset from the game Travian does not contain any psychological profiles for the individual players. |
| Demographics | At the time of data collection, the gaming provider did not collect demographic data at the individual level. Players were free to publish personal information in their player profiles. This information was freely available within the game but not verifiable for accuracy. Therefore, only the general demographic information listed in the data set is available for research purposes. |
| Communication medium | Players of the game Travian have the opportunity to communicate on an individual basis using the integrated e-mail-like communication system (IGM). Alternatively, they are free to use a forum that is available for the exchange of information within the alliance. However, this content was not available for data collection. In addition, some alliances chose to use external communication tools (e.g., messenger, team speak) that were also not traceable for research purposes. |
| Network-level variables | The existing dataset from the game Travian provides a variety of interactions and affiliation networks that can be used to construct network-based variables, as discussed above. The following transaction data is available for the construction of interaction networks: (1) communication, (2) transfer of resources, and (3) sending reinforcement (troops). Furthermore, the game provides additional data in the area of affiliation networks: (1) alliance membership, (2) mutual participation in armed conflicts (e.g., wars), (3) participation in joint construction projects (e.g., wonder of the world), and (4) account sitting (e.g., trust). |

networks was the fact that the content of the communication was not available to us. This data had been removed by the game operator to comply with data protection regulations. Thus, without this information, we could not distinguish between meaningful communication and chatter.

## Dependent variable

We chose *team performance* as the *dependent variable* based on a comparative time-sensitive ranking. The reason for using this measurement was that it is frequently (20.7%) studied in the area of network-based team research [41] and therefore, best suited for comparing results from offline research settings with our specific online world. For this, the MMOG Travian offers a very convenient performance metric, which is integrated into the game interface and is visible to the users themselves. As described in the section above, this ranking makes it possible to see in real time where one's alliance currently stands in relation to the other alliances. To obtain comparable results over time, we replicated this ordinal ranking on a daily basis and converted it to percentiles, which indicate the percentage of other alliances that rank lower than one's own team at the same point in time. As the final step, we averaged the values over the entire 60-day period to obtain a measurement point for the total observation period. We calculated this data as follows:

Let $A_d$ denote the set of alliances on a given day $d$, and let $rank(a, d)$ denote the rank of an alliance $a \in A_d$ on day $d$ with respect to the total number of points (of all that alliance's players). Thus, this rank ranges from 1 (lowest alliance) to $|A_d|$ (highest alliance). Then, the

**Table 2. The world of Travian—Contextual and social architectural factors based on Williams' research agenda for virtual mapping and other tests.**

| | |
|---|---|
| World size | The number of active players inhabiting the virtual worlds of Travian is not fixed and changes over time. At the time of data collection, the technical limit of active players that a server cluster could handle was about 20,000 to 30,000 active players. Players are free to register and start the game any time they want. The data shows that the majority of players sign up within the first 30 days. Many of them fail relatively quickly and then drop out. As described in the data section, during the one-year period that the virtual world was online, a total of 82,564 players registered, with an average of 8,385 players active daily. |
| Persistence | The online worlds of the game Travian are organized in rounds that last about a year. During this time, the world is persistent and events are based on past events. |
| Competitive versus collaborative | The game Travian is competitive and cooperative at the same time. Cooperation is a crucial factor in the success of alliances. Competition plays an important role between alliances. This tendency becomes stronger the closer the game world approaches the endgame. |
| Role play | Role-playing elements play only a minor role in Travian. The main task of the players is to develop their villages, administer infrastructure, troops and trade, as well as to be in communication with the other players. Personalized avatars that represent the player in the virtual worlds do not exist in Travian. The focus is rather on the administration of one's own kingdom. This is done via the village and overview map, as well as table-based user interfaces. |
| Sandbox versus linear representation | Travian can clearly be characterized as a sandbox game. The game offers an artificial world in which players are placed within a clearly defined framework in which their actions and interactions determine what happens within a game round. In addition to providing the framework in which the players can operate, the game design focuses primarily on providing the players with the game objectives, which slightly differ according to the three phases of the game. |
| Interaction affordances | The game Travian offers players five distinct dimensions for possible interactions. These categories are: (1) communication, (2) trade, (3) military support, (4) attack, and (5) raid. |
| Costs of behavior | Since Travian is a multiplayer game with game rounds lasting approximately one year, the behavior of a player in the game world has long-lasting consequences. Unlike in other games, there is neither the possibility to return to a previous state of the game, nor the opportunity to restart the game round. |
| Local culture | The operator of the game Travian offers different versions of the game for different countries. Therefore, within a game world, players from the same culture usually play with each other. The version of the game used here is the German language version of the game, with players originating primarily from Germany, Austria, and Switzerland. |

percentage rank of alliance $a$ on day $d$, denoted $pctrank(a, d)$ is given by:

$$pctrank(a, d) = \frac{rank(a, d) - 1}{|A_d| - 1}$$

Hence, the lowest alliance will have a score of 0, whereas the highest alliance will have a score of 1.

For a given period of time $T$, the performance of alliance $a$ is the average of daily scores of that alliance during that period and given by:

$$performance(a, T) = \frac{1}{|T|} \sum_{d \in T} pctrank(a, d)$$

In our study, the time period is chosen such that $|T| = 60$ days.

## Observation period

The choice of the right observation period was crucial, taking into account two key aspects: (1) the challenge that alliances can also form and dissolve at any time and that (2) the game is designed in three specific phases: early game, mid-game and late game, which differ in the goals to be achieved. Each phase is designed to last approximately 100–120 days and is triggered by a special event in the virtual world. The first phase (*early game*) begins with the start of the server. The players' primary goal in this phase is to familiarize themselves with the game environment and the other players, improve their infrastructure, and team up. The second phase (*mid-game*) is the period where leadership, intra-team coordination, and diplomacy between teams becomes more and more important. This phase is triggered by a special event: the introduction of the *artifacts*, the prerequisite for participation in the final race, the construction of the *Wonder of the World*, the goal of the third and final phase (*late game*) of the game. When we were choosing the observation period, we opted for the second phase. The main argument for this selection was that a massive reorganization process takes place following the introduction of the artifacts (see Fig 3). Further, team dynamics play a crucial role in this second phase and we assumed that they would be strongly reflected in the observed network patterns.

We, therefore, included all teams that existed on day 110 (the date of the trigger event) or that were subsequently formed. The second key condition for inclusion in the dataset was that the team existed for the full 60-day period, which ensured that only stable alliances were included. We chose these observation periods of 60 days to be able to construct meaningful networks from the communication and support interactions. Within the game, interactions such as communication or support between team members do not occur every day, so longer observation periods are required. With this in mind, we decided to use a moving window, in

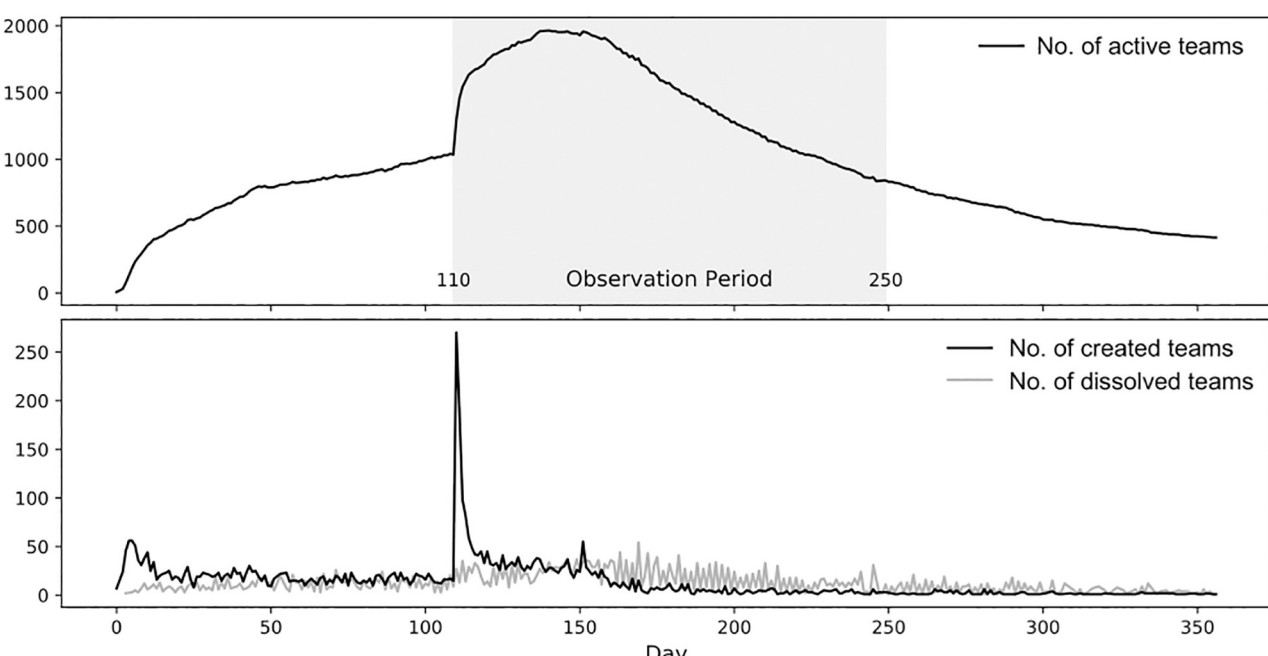

**Fig 3. Teams within Travian constantly form and dissolve.** In the first phase of the game, the number of alliance formations exceeds the number of dissolutions, which finds its peak at the beginning of the middle phase of the game. At this point a massive reorganization process takes place. In the later phases of the game, the number of team dissolutions exceeds the number of new formations, so the number of active teams steadily decreases.

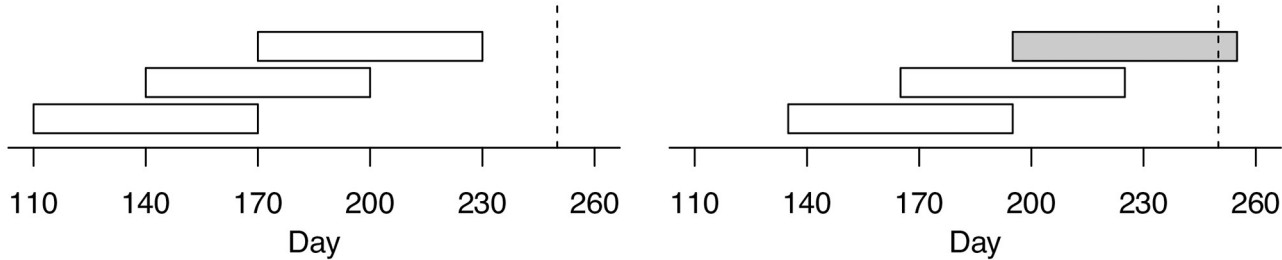

**Fig 4. A schematic representation of the observation periods taken from the mid game phase, starting with day 110 and ending with day 250.** Our modified moving window approach (advancing 30 days at a time) integrated all communication and support relationships that comprised a full 60-day period.

which we aggregated all communication and support relationships over 60 days, and also moved the window forward by 30 days at a time (see Fig 4). This decision implied that we allowed multiple observations (periods) per team, which allowed us to increase the sample size and also better reflect the context in which the team was at the current time. Finally, we defined day 250 (the beginning of the third phase) as the final closing day. This means that all observation periods we used had to be completed by that day.

## Statistical tests

The statistical methods applied in the various offline studies, which we referred to in the theoretical section, range from traditional OLS regressions to ERGMs, SIENA models, and simple logistic regressions. Most of the studies mentioned above limited themselves to examining only the impact of one or two network measures on performance. Since we present a collection of a total of nine different network metrics (and one control variable) in our exemplary case study, we had to find a methodology for the analysis that was compatible with the larger number of explanatory variables. The first and most critical challenge we faced in handling these network metrics was the extremely high inter-correlation between the independent variables. In addition, we faced the restriction that the distribution of our data did not meet the assumption of normal distribution. In the end, the first limitation prevented the application of multiple regression analysis to test our hypotheses. Although the bootstrapping of the data was helpful in overcoming the issue of the violated assumption of normality, very high VIF values prevented a meaningful use of multiple linear regression analysis. The use of structural equation models to determine the interplay of individual network measures and their influence on group performance was also not an alternative, as we could not draw on any theoretical foundation (models) for this purpose. After ruling out these first two alternatives, we decided to evaluate the influence of each network variable on performance independent from each other. For this, we considered two options. The first option was to apply Spearman's rank-order correlation, which would allow us to account for the ordinal nature of our performance data and whose transformation approach makes it more powerful in the context of non-normality than Person's coefficient [104]. The second option was to apply linear regression for each independent variable separately. This would have the benefit of including the (only available) control variable group size (N) in the analysis. After calculating and evaluating the two options, we found that group size only exerted minimal influence in the different samples (small, medium, large). In addition, the results of the Spearman correlation analysis were essentially consistent with those of the bivariate regression analyses. This insight led us to opt for excluding the group size variable from the analysis and to opt for Spearman's rank order coefficient due to its better ability to map ordinal data. In the following section, we will present the results in the

**Table 3. Correlation of each network measure with team performance for the groups of small- (15–25 members), mid- (35–45 members), and large-size teams (55–65 members) as well as all teams (10–143 members) (\*: $p \leq$ .01, \*\*: $p \leq$ .001).**

| Network | Feature | Small | Medium | Large | All |
|---|---|---|---|---|---|
| | Sample Size | 175 | 100 | 77 | 1179 |
| | Group Size (N) | 0.036 | 0.086 | 0.132 | 0.757 |
| Communication | Density | 0.435\*\* | 0.656\*\* | 0.830\*\* | -0.139\*\* |
| | Transitivity | 0.410\*\* | 0.574\*\* | 0.696\*\* | 0.476\*\* |
| | Centralization (betweenness) | 0.211\* | -0.160 | -0.149 | 0.281\*\* |
| | K-Core (relative size) | 0.319\*\* | 0.607\*\* | 0.676\*\* | -0.368\*\* |
| | K-Core (max. value) | 0.384\*\* | 0.576\*\* | 0.750\*\* | 0.784\*\* |
| | Group Centrality (degree) | 0.400\*\* | 0.537\*\* | 0.720\*\* | 0.673\*\* |
| Support | Participation | 0.668\*\* | 0.725\*\* | 0.835\*\* | 0.388\*\* |
| | Transitivity | 0.400\*\* | 0.563\*\* | 0.436\*\* | 0.579\*\* |
| | Centralization (in-degree) | 0.553\*\* | 0.644\*\* | 0.691\*\* | 0.248\*\* |

form of correlation tables (Table 3) combined with scatter plots (Figs 5 and 6), which are primarily intended to provide a better understanding of the context and the specifics of the data.

## Results

Table 3 presents our findings based on the four defined subgroups (small teams, mid-sized teams, large teams, and all teams). The largest sample, representing all team sizes without any clustering, contains 1,179 observations. However, in this group the unintended bias between

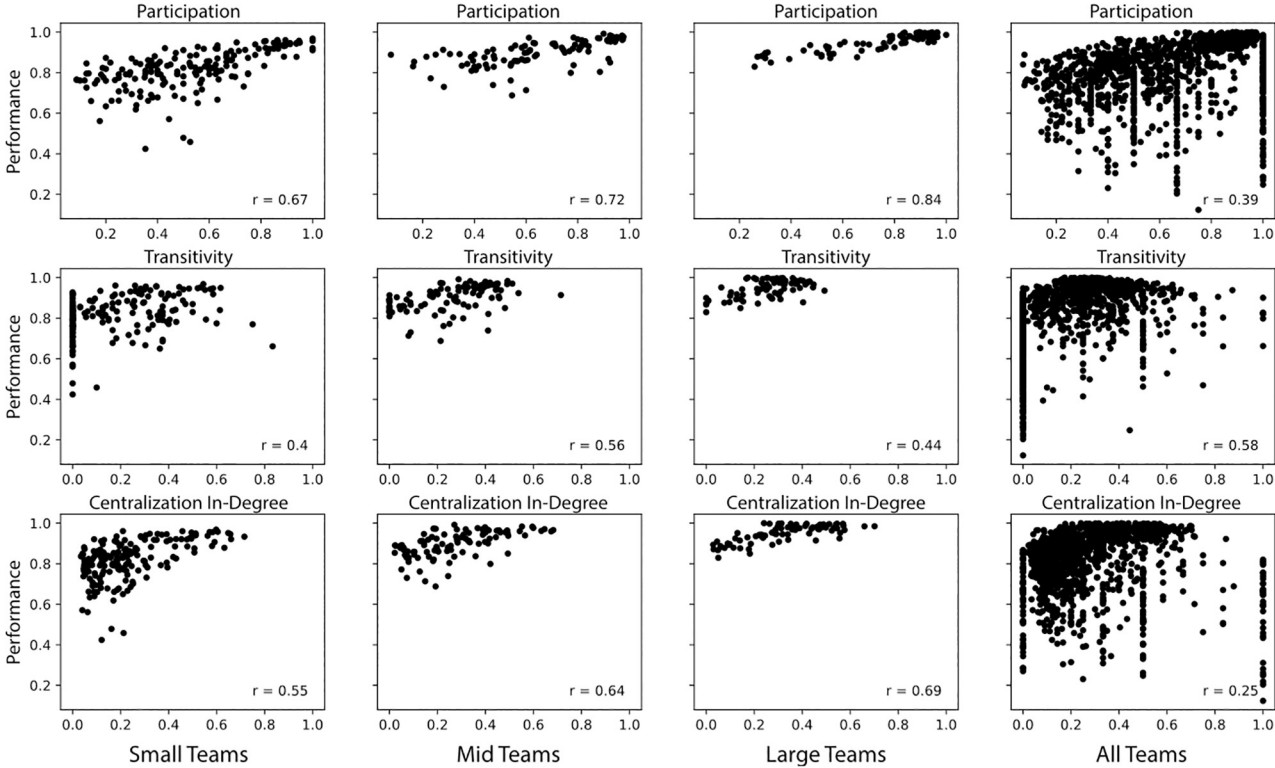

**Fig 5. Support networks.** Correlation of each measure with team performance for the groups of small-, mid-, and large-size teams as well as all teams.

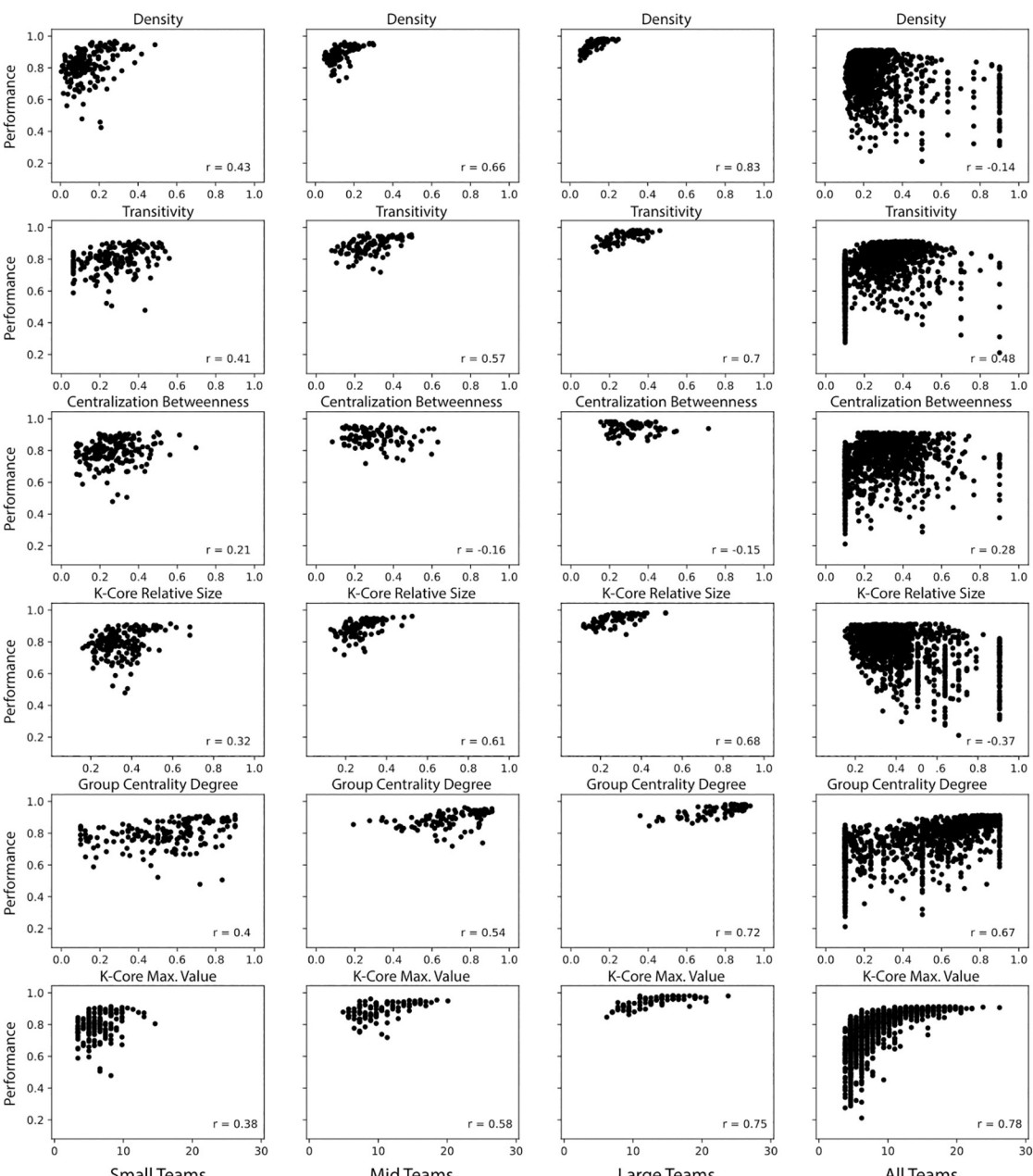

**Fig 6. Communication networks.** Correlation of each measure with team performance for the groups of small-, mid-, large-size teams as well as all teams.

group size and team performance is 0.757 (Spearman), which is a very high correlation. In contrast, the correlation (bias) in each of the three subgroups (small, medium, large) is much smaller but comes at the price of a much smaller sample size. Almost all results are significant or even highly significant. Further, it can be said that the observed values tend to be very stable across all four groups. However, this is different for some metrics from communication networks, particularly in density, betweenness centralization, and relative size of the k-core.

There are some clear tendencies to be found. With the higher average group size of the sub-sample, effect sizes increase for all cohesion-based features (participation and density) and

three communication-based leadership-related features (max. k-core, k-core relative size, group centrality). The same was true for the indicator of coordinated support activities (in-degree centralization) and transitivity in communication. Transitivity in support activities showed a stable pattern across all observed subgroups, with no observable trend based on average group sizes. Interestingly, a different - but not significant - pattern can be observed with respect to the impact of betweenness centralization in communication networks. While a higher centralization seems to have a positive effect for small groups, the effect changes from positive to negative for larger group sizes. However, it must be pointed out that the latter effect is nonsignificant and can therefore at best be regarded as an indication for future examinations. The trends represented by the correlations can also be found in the scatter plots in Figs 5 and 6.

Regarding our hypotheses, which we established in the theoretical part of this paper, it can be said that our results predominantly confirm them. Nonetheless, a deeper look is needed here. In both networks, higher levels of team cohesion (density and participation), as well as higher levels of transitivity, had a positive effect on team performance. Therefore, our results were able to confirm Hypotheses 1 (density), and 2 (transitivity) in communication networks, as well as Hypotheses 7 (participation), and 8 (transitivity) in support networks for all three subgroups. The fact that the correlations with team performance showed a negative sign for density in communication networks and a significantly lower one for participation in support networks can be explained by the fact that larger groups tend to have lower density. As mentioned in the theoretical section above, this is due to the way density is calculated. Combined with the existing high correlation between group size (N) and team performance, it is obvious that this effect changes the results significantly when all teams of different sizes are combined, so this result should be regarded as an artificial outcome without a deeper meaning. A more detailed look is needed regarding the outcome for Hypotheses 4 (k-core relative size). Here the results are contradictory and the results in the aggregated group (all sizes) do not confirm the hypotheses. This should not be overstated, as we have shown the importance of treating groups of different sizes differently and therefore these results are of limited applicability. The remaining Hypotheses 5 (group level centrality), 6 (k-core max. value), and 9 (coordinated actions) were fully confirmed by our results. Thus, it can be summarized that with one exception (Hypothesis 3), which should be further explored in future work, the results from our online study are consistent with previous findings from various traditional offline studies. Therefore, returning to our initial question, our results indicate that the social dynamics in the offline world turn out to be the same as in our specific online MMOG setting and that the *mapping test* was therefore successful. The next section leads us to the discussion and interpretation of our results.

## Discussion

Turning back to the introduction, the key question we addressed there was whether MMOGs have the potential to be the next "Petri dishes" or "supercolliders" for social scientists. Various works [1, 3, 6, 7, 10, 105] have addressed this question on an abstract level and left many questions unanswered. The idea of this study, therefore, was to show, based on a concrete proof of concept (case study), what such an exemplary research environment is capable of and where its limitations lie.

### Mapping test

An important aspect of this study was to conduct a mapping test, which was intended to provide us with further insight into whether human behavior in virtual worlds matches behavior

in the offline world. Methodologically, we followed the concept proposed by Williams [8]. It should be mentioned, however, that we were only able to partially implement this concept due to limited data availability for the control variables he proposed. Given this limitation in our study design, the mapping was successful and provides another example that confirms the initial thesis. This is further evidence that the behavioral patterns of people in traditional offline settings (e.g., work environments) and in newer online settings (e.g., online games) can be the same. Our conclusion from this is that in the end, it might not be so important to distinguish between online and offline worlds, especially in light of the fact that in today's world these boundaries are becoming increasingly blurred and that environmental conditions (context) are are also very different in the offline world. Future studies should, in our view, focus much more on the clear specification of environmental characteristics, which should increase the comparability of studies. This is not only relevant in the context of online environments, but a challenge that also affects traditional settings.

## Identified potentials of SNA-based research in MMOGs

In general, it can be said that our chosen combination of applying SNA-based team and leadership theory (our toolbox) to the MMOG Travian proved to be very effective. Using our exemplary research environment, we were able to show that MMOGs can provide different types of interactivity (multiple layers of networks) that allow us to look at the desired research question from different angles [106]. Furthermore, we used the example of communication networks to show that they not only map onto communication patterns in teams but can also tell us something about their (informal) leadership structures (based on cohesive subgroups) [34, 85, 107]. Beyond this general potential of MMOGs as a source of data, our case study also allowed us to identify a number of network metrics that have an impact on the success of teams. While we have been able to draw on a broad body of research on communication networks, knowledge about the influence of support networks within teams is still limited [42]. Therefore, we have limited ourselves here to basic features. The same applies to the influence of leadership, which we have shown can be mapped not only by established approaches (e.g., centralization measures applied to leadership networks) but also by analyzing communication patterns. It can be concluded that the network metrics we have presented (and others) offer great potential for further research into MMOGs for very different reasons. In particular, there is evidence that the greatest potentials of MMOGs as research environments lie in the following: (1) The availability of a variety of interaction networks between players provides data that allows the interdependencies between these dimensions to be examined (*Multiplexity*). (2) The descriptive use of network metrics allows a systematic investigation of whether the same behavioral patterns of actors can be found in online as well as offline worlds (*Mapping Tests*). (3) Time-stamped interaction data, widely available in MMOGs, makes it possible to address research questions that investigate the influence of time (*Longitudinal Approaches*). (4) Finally, the opportunities to study team and leadership dynamics and their influence on team performance in many ways.

## Identified limitations of SNA-based research in MMOGs

Despite the many new opportunities MMOGs offer, we should not ignore the limitations that these research environments face. Quite a number of arguments have already been discussed in the earlier theoretical work mentioned above [10, 28, 108]. Therefore, in the following, we would like to focus mainly on points that we encountered in our work with our particular Travian case study. The core issue we identified was that the incentive structures and environmental conditions (game design) found in these games do not necessarily match the study design

requirements. The following examples illustrate this in more detail. Typically, when planning a study, researchers determine exactly what data should be collected to best answer the research question [109]. This is different in the case of MMOG-based research. It is important to keep in mind that these games are mostly commercial projects, which often produce several million euros of revenue per month. This affects the priorities according to which the operators of the games act and according to which their software, databases, and infrastructure are designed. Game databases, therefore, do not necessarily store the data that researchers need to answer their research questions, but those that are relevant to the efficient live operation of the game. This includes that data is often stored for a very short time or calculated in real time. Regardless of its importance, much theoretically available data is not accessible to researchers at all or is only available at great expense. Therefore, researchers in this field often need some creativity and flexibility to work around these limitations. Likewise, regulation regarding data protection is increasingly posing a hurdle for data collection [110].

Two other major challenges are the incentive structures and implemented downside risks under which players, and thus study participants, interact within these games [8]. A key objective within the commercial games industry is to motivate players to participate in the game over the long term. The developers of a game are usually not interested in reproducing the offline world as best as possible, but in maximizing game fun, player retention, and revenues. Thus, it is important to evaluate already during the planning of the study whether the incentive structures and risks may be in conflict with the framework conditions that are necessary for the successful conduction of the study. These conflicts seem to be the rule rather than the exception.

The crucial question, therefore, is whether such biases can be resolved through prudent study design. In the case of our study, we faced the challenge of identifying the best-performing teams. The good news was that the game defined clear criteria on how to achieve this by providing a ranking. Unfortunately, it offered two different ways to achieve this goal: efficient team management or some other way, which has proven to be effective in the short term, simply by increasing the number of team members. This led to the problem that teams with similar levels of performance were not comparable because they differed in nature. Technically speaking, we were faced with a very strong correlation between team size and team performance, which turned out to be a major problem for conducting our study. The first argument against accepting this limitation is that comparing teams of different sizes makes little sense because they differ in their group dynamics (which we discussed in the previous section), which in turn is reflected in their structural patterns. Second, some network patterns we intended to use (e.g., density) respond very sensitively to an alternating number of nodes, which means that they can only be applied to groups of the same or similar size. Therefore, we decided to adapt our study design to the circumstances and avoid bias by defining subgroups based on group sizes (small, mid-sized, large). Those subgroups, which were now very homogeneous, showed a very low correlation with the number of nodes, which was an indication to us that we had been successful in eliminating the bias. However, as so often happens in such cases, we had to pay the price of having our sample size greatly reduced.

Another limitation we faced came from the fact that the game design allows members to join or leave an alliance at any time [6]. In contrast, the construction of an interaction network requires that the composition of the nodes remain unchanged over the observation period. We were unable to meet this condition with the game dataset we had, resulting in some blurriness in our data. This resulted in some contradictions, which we would like to describe in the following. Due to the fact that some alliance members joined or left the alliance within the 60-day observation period, some dyads existed for less than these 60 days, resulting in a lower

probability of being connected. In some extreme cases, there may even be the paradox that a connection between these two nodes is technically impossible because the two nodes were part of the alliances at different points in time. As a final limitation, we should mention that we had to make a couple of decisions regarding the time-based delineation (aggregation period, moving windows) and group classification (small, medium, large), which means that these decisions may have additionally influenced or even falsified the results of this study.

## Conclusion

Without a doubt, MMOGs have a lot to offer. In addition, we have not found any (unexplainable) objection in the literature or in our own work that the behavior of gamers in virtual environments is significantly different from human behavior in similar situations in the offline world. The crucial question is therefore why the great breakthrough of these "Virtual Laboratory Experiments" [3] did not materialize as expected a decade earlier. One reason for this might be that working with MMOGs as a research environment requires a very different mindset than traditionally found in the social sciences. Traditionally, researchers learn that the research question (RQ) defines the study design, which subsequently determines what data to collect. This is different in the work with MMOG environments. Here, data collection comes first. This is not a voluntary choice. In most cases, this can be taken as a given, with little room to change or expand the coverage of the data collected. If the researchers are lucky, they may have the opportunity to place surveys or special requests. In most cases, this will be too time-consuming for the game operator, so researchers will have to make the best of the secondary data provided. Another key issue is the challenges of applying *Big Data*. As is so often the case with large data sets, "[they] seldom can be taken off the shelf and used blindly in an 'as is' form" [111]. Rather, it requires the joint efforts of an interdisciplinary team to filter and process meaningful data from the vast amount of random data. In addition, a deep understanding of the data and a broad background knowledge of meaning and context is required. It is, therefore, no coincidence that most studies conducted with MMOG data come from a field where a high level of data handling expertise can be found. A Scopus database search using the keyword "MMOG" returned 683 results, with 528 coming from computer science and 125 were coming from the field of social sciences. Only 70 studies were assigned to both disciplines, which can be taken as an indication of interdisciplinary work. This brief example shows that studies that bring together the skills of computer scientists with social science issues are unfortunately still the exception, despite their great potential, and therefore the research results fall well short of the available potential. Finally, researchers should be aware of the game's incentive structures and how they reward or punish risk-taking. These different characteristics and nuances easily influence the social dynamics between players [25], and therefore it is an essential task for researchers to verify whether these environmental factors of the game change the "natural behavior" of players. For this, the researchers must know the game very well and in the end, this question also decides whether the research results can be applied to offline-world settings.

Given all this, we conclude that the virtuality of a research environment does not have to be something alien to reality, but is ultimately one aspect among many that must be taken into account. Further, *Organizational Network Analysis* offers a very promising toolbox for future research into Massively Multiplayer Online Games, as "networks can be more easily built from virtual world data than from real-world data" [8]. Undoubtedly, hurdles exist, but it is equally possible to overcome them through more interdisciplinarity, creativity, and a certain flexibility in finding new ways.

## Supporting information

**S1 File. Data used to compute directed communication and support networks.**
(ZIP)

## Acknowledgments

We would especially like to thank Travian Games GmbH, without their support in providing the data, the present research project would not have been feasible.

## Author Contributions

**Conceptualization:** Siegfried Müller, Jürgen Pfeffer.

**Data curation:** Siegfried Müller, Raji Ghawi.

**Formal analysis:** Raji Ghawi.

**Investigation:** Siegfried Müller.

**Methodology:** Siegfried Müller, Jürgen Pfeffer.

**Project administration:** Siegfried Müller.

**Software:** Siegfried Müller.

**Supervision:** Jürgen Pfeffer.

**Validation:** Siegfried Müller, Jürgen Pfeffer.

**Visualization:** Siegfried Müller, Raji Ghawi, Jürgen Pfeffer.

**Writing – original draft:** Siegfried Müller, Raji Ghawi.

**Writing – review & editing:** Siegfried Müller, Jürgen Pfeffer.

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
