## [Decision Letter · Decision Letter 0]

4 Apr 2022

PONE-D-22-03432Reviewing the potentials of MMOGs as research environments: A case study from the strategy game Travian.PLOS ONE

Dear Dr. Müller,

Thank you for submitting your manuscript to PLOS ONE. After careful consideration, we feel that it has merit but does not fully meet PLOS ONE’s publication criteria as it currently stands. Therefore, we invite you to submit a revised version of the manuscript that addresses the points raised during the review process. Please focus on generalization of results and better address used framework. Statistical analysis requires clarification to address comments from second Reviewer.

We look forward to receiving your revised manuscript.

Kind regards,

Jarosław Jankowski

Academic Editor

PLOS ONE

Journal Requirements:

2. PLOS ONE has specific requirements for studies using personal data from third-party sources, including social media, blogs, other internet sources, and phone companies (https://journals.plos.org/plosone/s/submission-guidelines#loc-personal-data-from-third-party-sources). These requirements include confirming data are collected and used in accordance with the company or website’s Terms and Conditions, obtaining appropriate ethics or data protection body review, and ensuring appropriate consent from individuals whose data are used in research. In this case, please ensure that your Ethics statement is in compliance with guidelines, and that you have complied with the company's (i.e., Travian's) Terms and Conditions, with appropriate permissions.

5. Please ensure that you refer to Figure 2 in your text as, if accepted, production will need this reference to link the reader to the figure.

Reviewers' comments:

Reviewer's Responses to Questions

**Comments to the Author**

1. Is the manuscript technically sound, and do the data support the conclusions?

Reviewer #1: Partly

Reviewer #2: Yes

2. Has the statistical analysis been performed appropriately and rigorously? 

Reviewer #1: No

Reviewer #2: I Don't Know

3. Have the authors made all data underlying the findings in their manuscript fully available?

Reviewer #1: Yes

Reviewer #2: Yes

4. Is the manuscript presented in an intelligible fashion and written in standard English?

Reviewer #1: No

Reviewer #2: Yes

5. Review Comments to the Author

Reviewer #1: To contribute to the ongoing debate of the utility of virtual worlds in representing real world scenarios, this study provides an overview of the findings from network-based team and leadership research, followed by an Massively Multiplayer Online Game (MMOG) case study attempting to replicate these findings.

While this area of study is certainly needful in light of steep advancements in the development and use of virtual platforms, unaddressed statistically concerns, make it difficult for these findings to be generalized. A key area of concern in the literature, and as highlighted by the authors to be the contribution of the current study, relates to the need for a mapping principle and framework, so as to aid the generalizability of results. However, the statistical approach of the study does not address this framework, failing to account for the influence associated with the research environment, a key tenet of the framework of which the study is primarily based on (see points #13- 15).

Further, while findings from past literature are generally well-integrated, the authors should concise points made by these various sources instead of populating paragraphs with direct quotes. This significantly reduces the readability nor is it entirely appropriate.

1) Page 2, line 9

“Williams” is not cited here, as it should be.

2) Page 2, line 18-21

It is not made explicit how unconsciously responding to media relates to the brain having evolved before the existence of media.

3) Page 3, line 30-31

While the previous examples supposedly illustrates how real-world behaviours may be mapped onto a virtual setting, the introduction of this point about machine learning utility in identifying interactions between network patterns and team performance is abrupt.

The authors may wish to remove this point or reposition it to a more appropriate location within the introduction.

4) Page 4, line 91,

The preliminary introduction of the acronym SNA needs to be first spelt out.

5) Page 4, line 116,

The citation for “Burt” should be denoted within this sentence.

6) Page 5, line 157,

The comma after formal networks appear to be redundant.

7) Page 5 line 171

While this line is aimed at introducing readers to the context under which a team operates, the link to “opportunities and constraints” are not immediately apparent. As such, authors may just chose to do without “(e.g., opportunities and constraints)”. The authors may wish to instead include relevant sub-headers between the parentheses, for example, “(i.e., density, transitivity, leadership patterns)”.

8) Page 8, line 180; page 9, 324; page 9, 354

While centrality and k-core are sub-sections under “Leadership Patters in Communication Networks”, the subsequent headings do not suggest this. Changes in the formatting of “Individual Level Centrality”, “K-core” and “Group Level Centrality” headings will improve readability.

9) Page 8, line 289

The location of the citation appears to be out of place.

10) Page 8, line 296

There appears to be a duplicate citation.

11) Page 12 ,line 469

The authors should justify why Travian was chosen.

12) Page 14, line 570-574

The authors may wish to align subsequent headers in accordance to the order of the framework explicated here to improve readability. Alternatively, the author may wish to reorder the framework here.

13) Page 18, line 741-744

It is mentioned here that items in the framework may be found under the section of “Research Settings (MMOG Travian)” they is no such section heading.

14) Page 18, line 740-741

The authors mentioned the importance of taking into account these contextual factors of the research environment, however, it is not explicated how these contextual factors were statistically accounted for.

15) Page 19, line 754-757

Given that factors relating to the research environment should be properly accounted for, it is unclear how Spearman’s rank-order correlation would be the best suited option, even with respect to concerns of the strong correlation between independent variables and non-normal distribution.

To explicate, it is not clear if the independent variables necessarily have to be examined concurrently with respect to team performance. In other words, why not transforming the data and using a linear regression for each independent variable? Or instead, why not structural equation modelling and allowing the independent variables to covary? Both these statistical methods allow for the effects of control variables to be accounted for.

16) Page 21, line 830-835

The authors initially appropriately highlighted that while mapping was successful in their study, this may not translate to all virtual environments (line 830-833). However, the authors proceed to make the conclusion that it might not be important to distinguish between online and offline worlds, directly contradicting the previous statement.

Reviewer #2: Referee report for PONE-D-22-03432 ‘Reviewing the potentials of MMOGs as research environments’

This clearly written paper uses data obtained from the Travian MMOG to test research hypotheses based on the findings of real-world studies of organisational effectiveness and leadership. These hypotheses boil down to testable associations between measures of organisational performance and measures which capture the structure of complex networks. The authors provide a comprehensive review of the real-world literature on network organisation, an explanation of the statistics used to measure network structure, an analysis based on the Travian data obtained by the authors, and a discussion of the results. Bar the points below, I feel this work potentially makes a valuable contribution.

Major comments

1. I was not particularly convinced that major contribution is showing that MMOG networks are just like real-work networks. Is that of substantive importance? The paper could just as easily be presented as an application of social network techniques to the behaviour of leadership/organisational behaviour in MMOGs, based on a substantive data example. Is that the patterns found loosely resemble those found in real-world networks simply to do with that involvement of human actors and the stakes/context their interactions take place in?

2. The paper is clearly written but does feel repetitive. Could the definitions of centrality, k-core, etc., which appear on pages 8-10 and again 15-16 be merged? I personally would have preferred the concepts to be introduce alongside the descriptions of the statistics, because I found the concepts to be hard to follow in the abstract without any graphical schematic or equation to follow (especially k-core: what is a maximal cohesive subgroup, the hierarchical structure of the k and k+1 cores, etc.). Doing this would also reduce the sense of repetition and prevent readers from forgetting too many details from earlier.

3. Statistical analysis: I am a little surprised that multicollinearity prevented a regression analysis from being performed. The sample sizes are small but not too small. We are thus prevented from understanding the partial effects of each network measure (i.e. those of a per-unit increase with the other measures held fixed) on the performance outcome, and must rely on bivariate effects instead. Non-normality could have been dealt with using a nonparametric bootstrap of one form or another. I would ideally like to see the regression analysis results or, at least, more justification as to why a regression analysis was not performed, because the reasoning currently given (pages 18-19) are vague.

Minor comments

4. From page 5: I would like a clearer, if brief, statement about the importance of communication and support for the study of leadership and performance. The importance emerges the more one reads (especiallyon page 14), but it would be helpful to flag the key reasons earlier.

5. Page 12, discussion of alliances. A simple worked example linked to a graphical representation would help the reader no end in understanding how these alliances are formed and how each works together and its overall performance measured.

6. Table 1: To clarify, there are 352 teams (of small, medium and large sizes) and the total number of individuals in these teams is 1179, but each person is in one team? Or is it that there are 352 teams and 1179 – 252 single-person teams? If the former, this needs to be made clearer in the table because it is confusing to mix sums of different unit sizes.

6. PLOS authors have the option to publish the peer review history of their article (what does this mean?). If published, this will include your full peer review and any attached files.

Reviewer #1: No

Reviewer #2: No

---

## [Author Response · Author response to Decision Letter 0]

1 Aug 2022

Dear Reviewers,

Thank you very much for your feedback on our paper. Your comments were very helpful in

improving the quality of the work.

In the attached file you will find an overview of the changes we made compared to the original version.

Kind regard,

Siegfried Müller

---

## [Decision Letter · Decision Letter 1]

2 Jan 2023

PONE-D-22-03432R1Reviewing the potentials of MMOGs as research environments: A case study from the strategy game Travian.PLOS ONE

Dear Dr. Müller,

Thank you for submitting your manuscript to PLOS ONE. After careful consideration, we feel that it has merit but does not fully meet PLOS ONE’s publication criteria as it currently stands. Therefore, we invite you to submit a revised version of the manuscript that addresses the points raised during the review process.

 As editor, I would require two changes for acceptance.  First, in your materials and methods section you state that the maximum team size is 60, but the number of individuals in a team in your dataset goes up to 145.  The discrepancy, which may be due to the fact that members can leave alliances and be replaced by new members, needs to be explained.  (If you did explain it and I missed it, please point me to where you explained it.)  Second, in your results section you state that higher betweenness leads to lower performance for medium and large teams.  But your table indicates that these relationships are nonsignificant.    

I do have three comments, all of them for future reference.  First, this manuscript is quite long.  While PLOS does not have any length limitations, I'm concerned that readers may simply fail to finish it or skip over much of it. Second, you may already know this, but the reason that you found collinearity between the social network variables is that network density influences other network structures; this is the reason it's necessary to control for the number of edges in an ERGM model.  You can also expect centrality measures to be collinear.  Third, if you have the time stamps it might be interesting to pull out a subsample of the networks and look at their evolution using a longitudinal model such as RSiena or TERGM.

We look forward to receiving your revised manuscript.

Kind regards,

Keith Leverett Warren, Ph.D.

Academic Editor

PLOS ONE

Journal Requirements:

Reviewers' comments:

Reviewer's Responses to Questions

**Comments to the Author**

1. If the authors have adequately addressed your comments raised in a previous round of review and you feel that this manuscript is now acceptable for publication, you may indicate that here to bypass the “Comments to the Author” section, enter your conflict of interest statement in the “Confidential to Editor” section, and submit your "Accept" recommendation.

Reviewer #2: (No Response)

2. Is the manuscript technically sound, and do the data support the conclusions?

Reviewer #2: Yes

3. Has the statistical analysis been performed appropriately and rigorously? 

Reviewer #2: N/A

4. Have the authors made all data underlying the findings in their manuscript fully available?

Reviewer #2: Yes

5. Is the manuscript presented in an intelligible fashion and written in standard English?

Reviewer #2: Yes

6. Review Comments to the Author

Reviewer #2: My previous comments have been broadly addressed. I'm disappointed that the authors chose not, as per my suggestion, to use graphical illustrations to demonstrate some of the concepts from social network analysis. Only major typo I spotted was "betweenness" in the abstract.

7. PLOS authors have the option to publish the peer review history of their article (what does this mean?). If published, this will include your full peer review and any attached files.

Reviewer #2: No

---

## [Author Response · Author response to Decision Letter 1]

2 Jan 2023

Dear Keith Leverett Warren,

Thank you very much for your final feedback on our paper. Your additional comments are appreciated and are helpful in improving the quality of our work.

In the attachted file you will find an overview of the changes we made compared to the previous version.

Kind regard,

Siegfried Müller

---

## [Editor Report · Decision Letter 2]

17 Jan 2023

Reviewing the potentials of MMOGs as research environments: A case study from the strategy game Travian.

PONE-D-22-03432R2

Dear Dr. Müller,

We’re pleased to inform you that your manuscript has been judged scientifically suitable for publication and will be formally accepted for publication once it meets all outstanding technical requirements.

Kind regards,

Keith Leverett Warren, Ph.D.

Academic Editor

PLOS ONE

Additional Editor Comments (optional):

Thanks for your patience!
---

## [Editor Report · Acceptance letter]

19 Jan 2023

PONE-D-22-03432R2 

Reviewing the potentials of MMOGs as research environments: A case study from the strategy game Travian. 

Dear Dr. Müller:

I'm pleased to inform you that your manuscript has been deemed suitable for publication in PLOS ONE. Congratulations! Your manuscript is now with our production department. 

Kind regards, 

on behalf of

Dr. Keith Leverett Warren 

Academic Editor

PLOS ONE